# Structural comparison of GLUT1 to GLUT3 reveal transport regulation mechanism in sugar porter family

Tânia Filipa Custódio[1],*, Peter Aasted Paulsen[1],*, Kelly May Frain[1], Bjørn Panyella Pedersen[1,2] (ID)

The human glucose transporters GLUT1 and GLUT3 have a central role in glucose uptake as canonical members of the Sugar Porter (SP) family. GLUT1 and GLUT3 share a fully conserved substrate-binding site with identical substrate coordination, but differ significantly in transport affinity in line with their physiological function. Here, we present a 2.4 Å crystal structure of GLUT1 in an inward open conformation and compare it with GLUT3 using both structural and functional data. Our work shows that interactions between a cytosolic "SP motif" and a conserved "A motif" stabilize the outward conformational state and increases substrate apparent affinity. Furthermore, we identify a previously undescribed $Cl^-$ ion site in GLUT1 and an endofacial lipid/glucose binding site which modulate GLUT kinetics. The results provide a possible explanation for the difference between GLUT1 and GLUT3 glucose affinity, imply a general model for the kinetic regulation in GLUTs and suggest a physiological function for the defining SP sequence motif in the SP family.

## Introduction

In humans, GLUT proteins are responsible for cellular glucose uptake. Basal cellular glucose uptake is mediated by GLUT1 (Mueckler et al, 1985), whereas GLUT3 is specifically found in neurons and other tissues with high energy demand (Simpson et al, 2008). In line with its physiological role, GLUT3 has been shown to have an increased glucose affinity (~3 mM) in comparison to GLUT1 (~10–20 mM) (Burant & Bell, 1992; Nishimura et al, 1993; Maher et al, 1996; Bentley et al, 2012; De Zutter et al, 2013); however, the affinity disparity cannot be explained by differences in their glucose binding sites, as they are structurally identical (Deng et al, 2014, 2015).

GLUTs belong to the Sugar Porter (SP) family, which as the largest branch of the Major Facilitator Superfamily (MFS), is found in all kingdoms of life (Baldwin, 1993). MFS proteins share a common fold comprising of 12 transmembrane helices (M1–M12) with a twofold pseudo-symmetry between the N-domain (M1-6) and the C-domain

(M7-12). They are also defined by a signature motif, the "A motif," with a consensus sequence of $Gx_3[D/E][R/K]xGx[R/K][K/R]$ (Nishimura et al, 1993). Due to the pseudo-symmetry, the A motif is found twice, located in the cytosolic loop connecting M2 and M3 of the N-domain and in the cytosolic loop connecting M8 and M9 of the C-domain. In GLUT1 the A motif takes the form $G_{84}LFVNRFGRR_{93}$ and $L_{325}FVVERAGRR_{334}$. The A motif is believed to be a key determinant of transport kinetics (Cain et al, 2000; Jiang et al, 2013; Nomura et al, 2015; Zhang et al, 2015), and it may also modulate transport by direct lipid interactions (Martens et al, 2018). Within the MFS superfamily, the SP family have a family-defining sequence, the "SP motif" located, also in duplicate, on both the cytosolic side of the N-domain directly after M6 and in the C-domain directly after M12. The SP motif takes the form of $P_{208}ESPR_{212}$ and $P_{453}ETKG_{457}$ in these two locations in the GLUT1 protein (Fig S1) (Baldwin, 1993). The importance of the SP motif has been demonstrated through mutational studies, and is highlighted by its strong conservation in the SP family (Seyfang & Landfear, 2000; Sun et al, 2012); however, the functional role of the SP motif has not yet been established.

SP proteins alternate between inward facing and outward facing conformations with respect to the central substrate binding site, and the transition between these two states define sugar transport (Fig 1A). When the transporter is in the outward facing conformation ($C_{out}$), extracellular sugar can bind to the central binding site ($C_{out}S$), and be transported into the cell ($C_{in}S$) as dictated by the concentration gradient across the membrane. The following conformational transition from the substrate-free inward conformation ($C_{in}$) to the substrate-free outward conformation ($C_{out}$) is needed to reset the transporter, and has been experimentally shown to be the rate-limiting step in GLUTs, consistent with thermodynamic models (Lowe & Walmsley, 1986; Wheeler & Whelan, 1988; Zhang & Han, 2016). During transport, the A motif has been found to control the degree of stabilization of the outward conformation (Jiang et al, 2013; Zhang et al, 2015).

GLUT1 and GLUT3 allow for a useful comparison to understand GLUT and SP protein kinetics, given their different kinetic properties, despite possessing identical substrate-binding pockets. However, few studies have directly compared the determinants of GLUT1 and GLUT3 kinetics (Burant & Bell, 1992; Maher et al, 1996).

---

[1]Department of Molecular Biology and Genetics, Aarhus University, Aarhus C, Denmark   [2]Aarhus Institute of Advanced Studies, Aarhus University, Aarhus C, Denmark

Correspondence: bpp@mbg.au.dk
*Tânia Filipa Custódio and Peter Aasted Paulsen contributed equally to this work

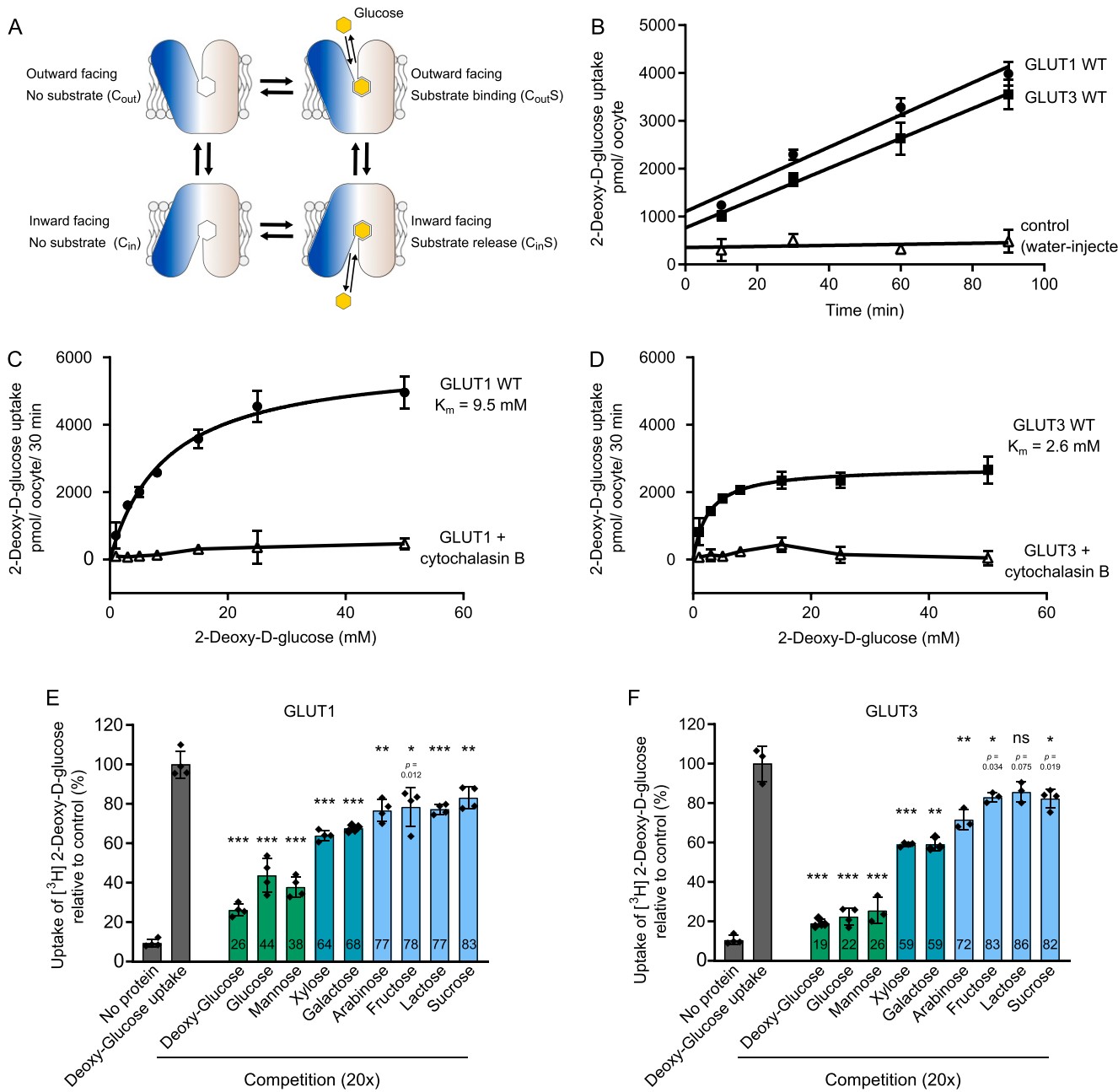

**Figure 1. Comparison between GLUT1 and GLUT3.**
**(A)** Schematic model for transport by GLUTs, alternating between two major conformations with the substrate-binding site exposed to the inside and outside of the cell. Transition between these conformations leads to sugar transport across the membrane following the substrate concentration gradient. **(B)** Uptake of 2-DG into GLUT1-injected *Xenopus* oocytes (circle), GLUT3-injected oocytes (squares) or water-injected oocytes (open triangles) at an initial outside concentration of 5 mM 2-DG. For both proteins, 2-DG uptake was linear in the range of 90 min. Data for all assays are mean ± SD of three or more replicate experiments. **(C)** Determination of the kinetic parameters for the transport of 2-DG of GLUT1. The data were fitted using the Michaelis–Menten non-linear fit, yielding a $K_m$ = 9.5 ± 1.0 mM and $V_{max}$ = 5,988 ± 226 pmol/oocyte/30 min. Sugar uptake was inhibited in GLUT1-injected oocytes exposed to cytochalasin B. **(D)** Determination of the kinetic parameters for the transport of 2-DG of GLUT3. The data were fitted using the Michaelis–Menten non-linear fit, yielding a $K_m$ = 2.6 ± 0.4 mM and $V_{max}$ = 2,731 ± 94 pmol/oocyte/30 min. Sugar uptake was inhibited in GLUT3-injected oocytes exposed to cytochalasin B. **(E)** Substrate selectivity of GLUT1 determined by competition assay in oocytes exposed to 5 mM 2-DG and 20× fold of the competing sugar, for 15 min. **(F)** Substrate selectivity of GLUT3 determined by competition assay in oocytes exposed to 5 mM 2-DG and 20× fold of the competing sugar, for 15 min. Data information: In (B, C, D, E, F) Data for all assays are mean ± SD of three or more replicate experiments. In (E, F) ns, Not significant; *$P$ ≤ 0.05; **$P$ ≤ 0.01; and ***$P$ ≤ 0.001 by $t$ test. $P$-value is shown for ns and *.

Large variations between published studies can be attributed to the assay type and substrate used to measure transport (discussed by Maher et al [1996]). It is also important to consider the protein/lipid environment in experiments, as GLUT proteins are highly sensitive to lipid compositions in the membrane (Wheeler et al, 1998; Hresko et al, 2016). A comparative study of the two proteins requires identical experimental setups, alongside high-resolution structural information. To this end, we report a 2.4 Å resolution structure of

human GLUT1 that reveals two previously unknown features: a chloride-ion binding site between the SP and A motifs of the N-domain, and an intracellular glucose and/or lipid-binding site at this SP motif. We present a direct biochemical comparison between GLUT1 and GLUT3 in identical experimental setups to examine and revalidate the differences between the two proteins. Based on the structural data, we pinpoint key elements of the cytosolic domain that can modulate transport kinetics. These elements reside within the A and SP motifs and they provide a functional framework to better understand the role of these motifs in all SP and MFS proteins. The results provide a provisional model which can explain the kinetic differences between GLUT1 and GLUT3, and furthermore suggests their transport regulation by a structural framework found in the SP motif that can interact with lipids and/or intracellular sugar. We support this new model by mutational analysis and identify a single point mutation in the SP motif that can convert GLUT1 into a transporter with GLUT3-like kinetics and vice-versa.

## Results

### Affinity and selectivity comparisons between GLUT1 and GLUT3

To directly compare the kinetics of human GLUT1 and GLUT3 we expressed both proteins in parallel in *Xenopus* oocytes and measured 2-deoxyglucose (2-DG) uptake (Fig 1B). Michaelis–Menten analysis of substrate saturation kinetics using GLUT1 result in a $K_m$ for 2-DG uptake of 9.5 mM (Fig 1C), comparable with previous reports in the literature (Burant & Bell, 1992; Bentley et al, 2012). The uptake is inhibited by the GLUT inhibitor cytochalasin B. In the same experimental setup, GLUT3 has a $K_m$ value of 2.6 mM and also displays cytochalasin B inhibition (Fig 1D). These kinetics are akin to the values obtained in rat primary cerebral granule neurons reported previously (Maher et al, 1996). Protein levels of GLUT1 and GLUT3 in the oocyte membranes were detected by Western blot using an identical engineered epitope (derived from GLUT1) for both proteins. The results show that expression levels are comparable (Fig S2A) and that kinetic parameters of GLUT3 are not affected by the introduced epitope (Fig S2B). We assess the $V_{max}$ levels per oocyte, and estimate the $V_{max}$ of GLUT1 to be 5,988 pmol/oocyte/30 min, and the $V_{max}$ of GLUT3 to be twofold lower at 2,731 pmol/oocyte/30 min (Fig 1C and D). Notably, in contrast to an older study that calculated a higher turnover for GLUT3 compared with GLUT1 (Maher et al, 1996), our results support a more recent study that suggest GLUT3 has a lower turnover than GLUT1 (De Zutter et al, 2013).

GLUT1 and GLUT3 substrate selectivity was screened in an identical setup to above with a competition assay. The takeaway message is that glucose and mannose strongly compete with 2-DG uptake in both transporters (Fig 1E and F). Other sugars also compete for uptake, but to a lesser extent, in accordance with previous literature (Burant & Bell, 1992; Deng et al, 2015). The similarities in selectivity are consistent with the structural conservation of two substrate binding sites and hence reinforce the idea that other regions of the proteins must be responsible for

conferring the difference in substrate affinity and transport capacity between GLUT1 and GLUT3.

### High resolution GLUT1 structure reveals a novel chloride binding site

We subsequently solved the crystal structure of human GLUT1 to 2.4 Å resolution (Rfree 22.9%) (Figs S3 and S4 and Table S1). The overall structure adopts an inward-open conformation that is almost identical to the 3.2 Å model published previously (RMSD$_{(CA)}$ 0.4 Å) (Fig S5A) (Deng et al, 2014). The structure consists of residues 9–455 (of 492 residues total) and contains the N-domain, the cytosolic ICH domain (ICH1-ICH4), and the C-domain (Fig 2A). The high quality of the electron density map, compared with earlier studies, allows for the identification of several new ligands and 13 water molecules in the structure, alongside well-defined side chains (Fig S4).

GLUT1 crystals were grown in the presence of glucose and the detergent Nonyl-$\beta$-D-Glucoside (NG), which has a glucose head-group. In our maps, we clearly observe an NG molecule in the central sugar-binding pocket in the N-domain (Figs 2A and B, S3, and S4), despite having a sixfold molar excess of glucose present (40 versus 6.5 mM). The entirety of the NG density is well defined, although there is less density for the tail, which could reflect its flexibility or that the site is occupied by a mixture of NG and glucose molecules. Indeed, occupancy refinement using a mixed ligand model yields a reproducible 40/60% ratio between glucose and NG. Residues from the C-domain coordinate the glucose headgroup, with seven polar interactions established through residues from M7, M8, and M10. When compared with the high resolution glucose bound GLUT3 structure (PDB 4ZW9, 1.5 Å) (Deng et al, 2015), we can confirm the atomic interactions are structurally identical between the proteins (Fig S5B). Next to the glucose headgroup, we also observe an additional tubular density close to the binding site that fits with a polyethylene glycol (PEG) molecule, derived from the experimental setup (Figs 2 and S4). The position of this PEG molecule matches previously identified inhibitor positions in GLUT1 (Fig S6).

The two SP motifs are located directly below the two A motifs in the N- and C-domain of GLUT1 (Fig 2C and D). Here, an unexpected density feature was present in the electron density map. There was a very strong spherical density (9.1 sigma peak) between the SP motif and the A motif of the N-domain, which could not be explained by a water molecule. The coordinating atoms indicate a negatively charged species such as a chloride ion, so we devised a follow-up experiment exploiting the strong anomalous scattering from bromide, a heavier chemical congener of chloride. By replacing Cl$^-$ with Br$^-$ in the crystal it is possible to search for anomalous density peaks to identify bromide ion positions which will reveal the original chloride position in the previous dataset (Ekberg et al, 2010). This experiment lead to a single strong anomalous peak (7.3 sigma) at the position of the spherical density in the native dataset, allowing us to unequivocally identify this peak as a Cl$^-$ ion (Fig 3) positioned between the SP and A motif in the N-domain.

The chloride ion is coordinated by several residues of the A-motif in the N-domain including the backbones of Arg92, Arg93, as well as Glu209 from the SP motif, with hydrogen bonds ranging in

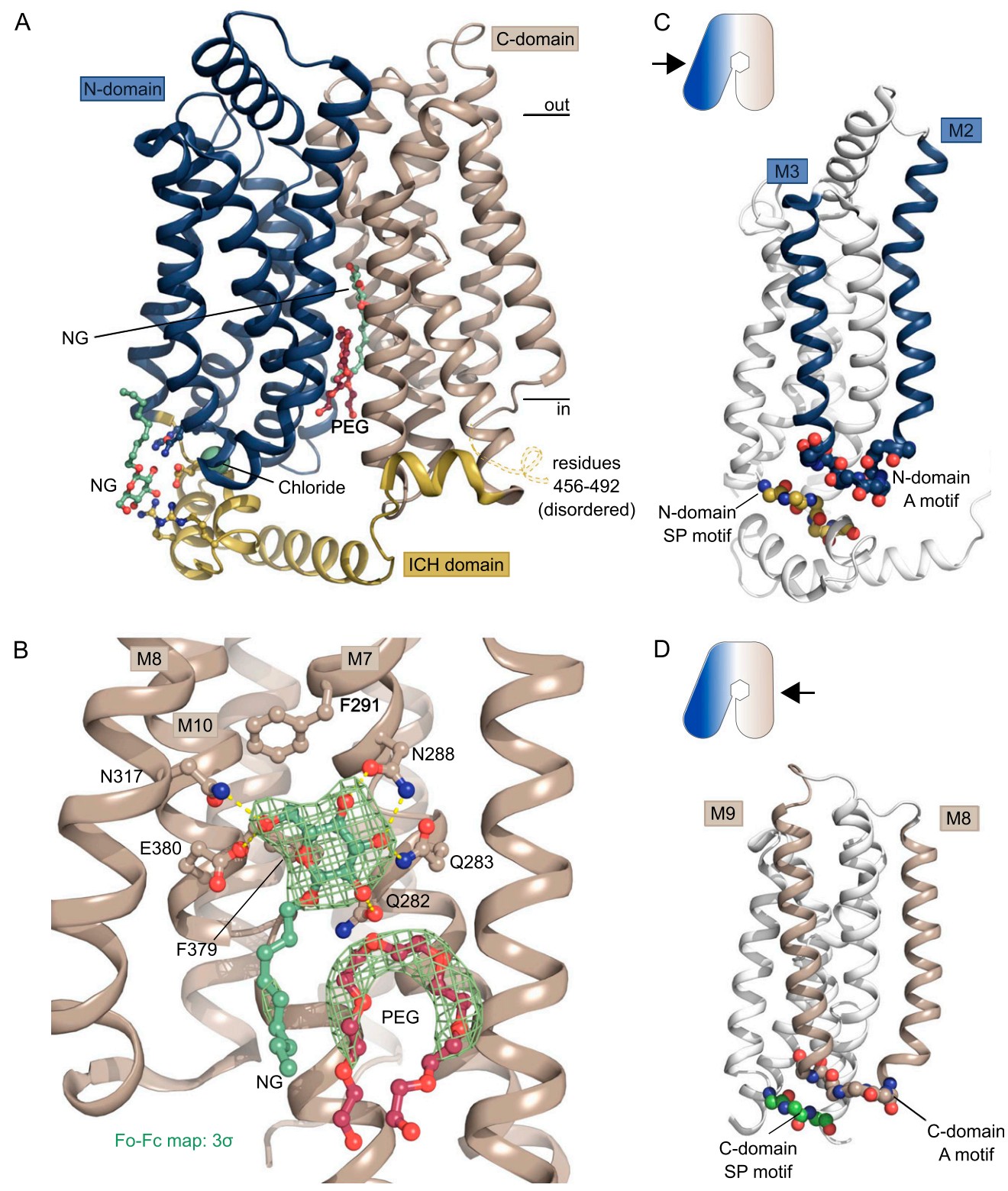

**Figure 2.   Crystal structure of GLUT1 reveals new ligands.**
**(A)** The overall structure of GLUT1 in the inward-open conformation. The structure represents a bound state with an NG molecule (shown as sticks) in the central cavity formed between the N-domain (blue) and the C-domain (brown), followed by a PEG molecule (shown as sticks). In close proximity with the ICH domain (yellow) another NG molecule (shown as sticks) was found, as well as a chloride ion (shown as a sphere). Selected residues are shown as sticks. Black lines depict the approximate location of the membrane. **(B)** Coordination of the glucose moiety in the central cavity by residues from C-domain. Hydrogen bonds are represented by yellow dashes (2.6–3.6 Å distances). The omit Fo-Fc density for NG and PEG is contoured in green at 3 σ. **(C)** Side view of GLUT1 shows the localization of the N-domain Sugar Porter motif directly underneath the A-motif from the M2-M3 loop. Signature motifs are shown as spheres. **(D)** Side view of GLUT1 shows the localization of the C-domain Sugar Porter motif directly underneath the A-motif from the M8-M9 loop. Signature motifs are shown as spheres.

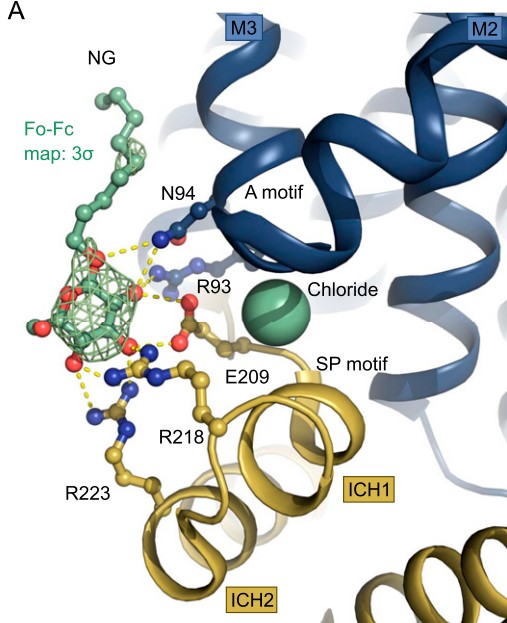

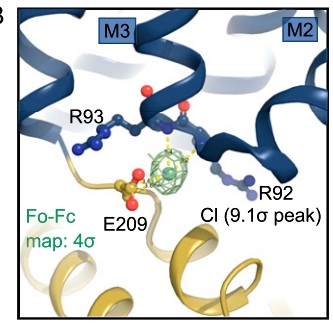

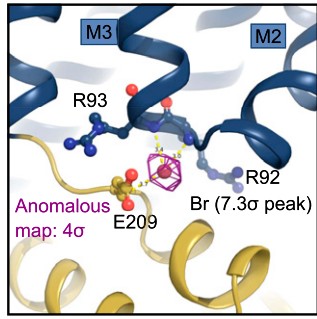

**Figure 3. Intracellular binding of Chloride and NG stabilize the inward-open conformation.**
**(A)** Intracellular NG is coordinated by residues from the Sugar Porter and A motifs. A chloride ion (shown as spheres) neutralizes the A-motif. Hydrogen bonds are represented by yellow dashes (2.6–3.6 Å distances). The omit Fo-Fc density for NG is contoured in green at 3 $\sigma$. **(B)** Coordination of the chloride ion by residues of the A-motif. The Fo-Fc electron density, shown in green mesh, is contoured at 4 $\sigma$ (top). The anomalous signal for the chemical chloride congener, bromide, shown in magenta mesh, is contoured at 4 $\sigma$ (bottom).

distance from 3.2 to 3.6 Å (Figs 3B and 4). Human intracellular chloride concentrations range from 4 to 70 mM, depending on cell type and measurement method (Berend et al, 2012), which is lower, but in a comparable range to the concentration in the experiment. The site could thus potentially be a crystallization artifact, but also might play a physiological role in the interplay of the SP and A motif in the SP protein family.

### The role of the SP motif found in the N-domain

The identification of a chloride binding site highlights a key change between the inward- and outward conformation found in the SP motif of the N-domain (Fig 4). All structurally characterized members in the SP family show a well-defined and precisely positioned conformation for the glutamate residue of the SP motif in the N-domain (E209 in GLUT1) (Fig S7). This essential glutamate interacts with the A motif in the outward conformation, but flips out and away from the A motif in the inward conformation. It is well established that residues of the A-motif stabilize GLUT proteins in an outward conformation, ready to receive glucose, via inter-TM salt bridges. Disruption of these networks triggers favorable formation of the inward state (Martens et al, 2018). When expanding the analysis to other SP structures available, a consistent picture emerges: In the outward-facing conformation, the glutamate residue of the human GLUT3 SP motif (Deng et al, 2015), rat GLUT5 (Nomura et al, 2015), and the *Arabidopsis* proton symporter STP10 (Paulsen et al, 2019) interacts with the A motif, establishing hydrogen bonds with amide groups from the last two residues (R92 and R93 in GLUT1) of the A motif (Fig S7A). We name this interaction network, found only in outward conformations of SP proteins, the "SP-A network."

In our inward-facing conformation of human GLUT1, alongside bovine GLUT5 (Nomura et al, 2015) and the bacterial sugar/proton

symporters XylE (Wisedchaisri et al, 2014) and GlcPse (Iancu et al, 2013), the glutamate residue is pointing away from the A motif and towards the cytosol (Fig S7B). The position of the glutamate acid group in the SP-A network is replaced by the identified Cl⁻ ion in the GLUT1 structure (Fig 4A). Notably, a similar strong peak is observed in the same site in the bovine GLUT5 electron density map. This suggests a mechanism where the glutamate residue in the SP-A network competes with a Cl⁻ ion for interaction with the A motif, whereas the glutamate interaction to the A motif stabilizes the outward conformation.

We set forth to explore our hypothesis by removing the acid group of the relevant glutamates of both GLUT1 (E209Q) and GLUT3 (E207Q) (Fig 4). These mutations should weaken the interaction with the A motif in the outward conformation. As a result, the outward-facing conformation should be disfavored and transport affinity should correspondingly decrease, as the protein will stabilize in the inward conformation. Oocyte uptake revealed a reduced 2-DG uptake because of a ~2-fold decrease in apparent binding affinity compared with the wild-type protein (Fig 4B and C). According with our model, $V_{max}$ values, which can be defined as the frequency of the interconversion between the outward and the inward conformations, should decrease. This effect is seen using the GLUT3 (E207) mutant but not the GLUT1 (E209Q) mutant. The Western blot did show protein expression levels were higher for the mutants compared with WT proteins (Fig S8), restricting our $V_{max}$ analysis. However, both mutations support our hypothesis that the role of the glutamate's acidic group is to interact with the A motif to stabilize the outward conformation.

### Ligand interaction with the SP motif of the N-domain

We observe an additional density on the cytosolic side of GLUT1 interacting with the N-domain SP motif. The main part of this

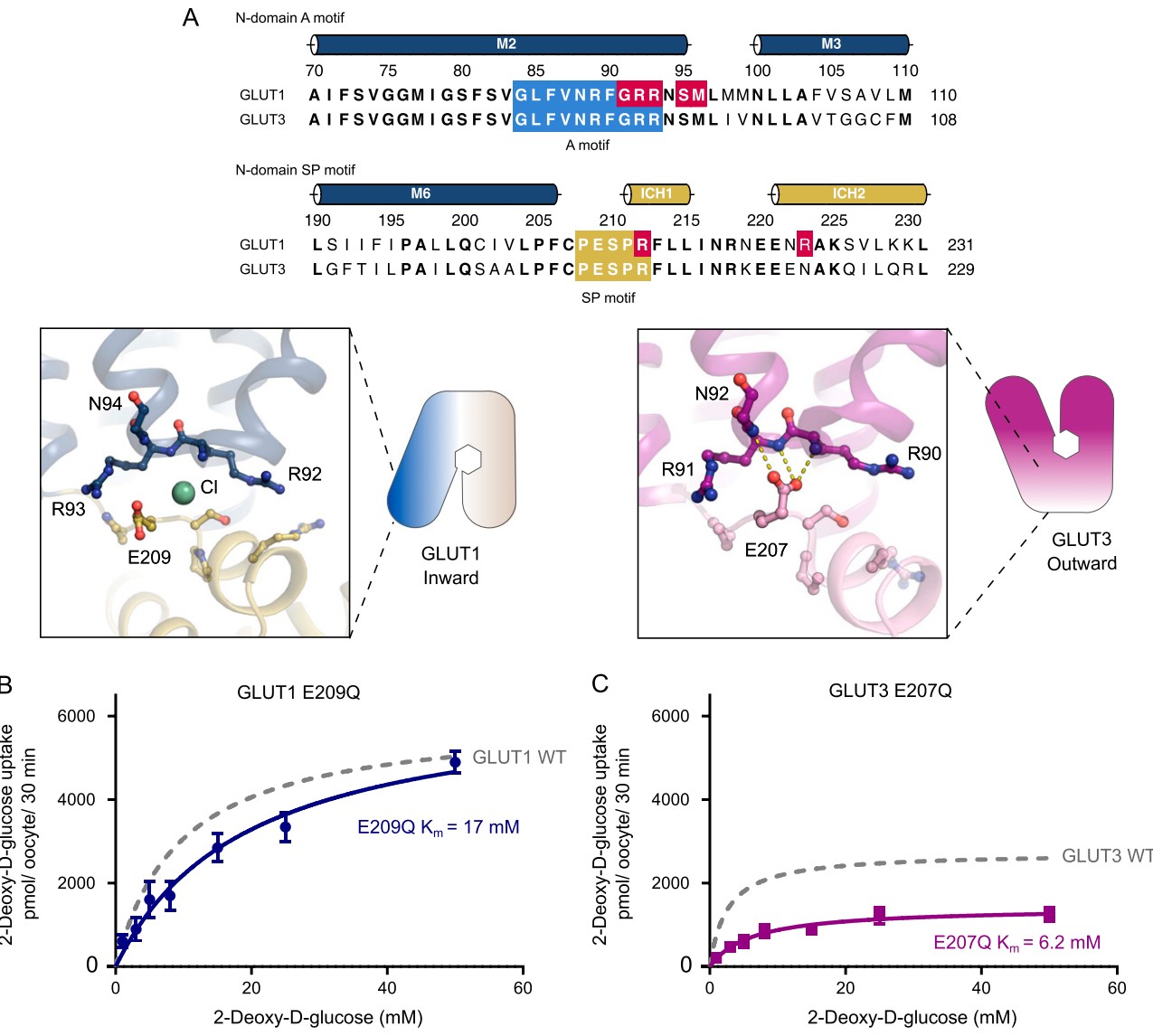

**Figure 4. The N-domain Sugar Porter (SP)-A network between inward and outward states.**
**(A)** Sequence alignment between GLUT1 and GLUT3 of the N-domain A motif (blue) and the N-domain SP motif (yellow). Residues involved in GLUT1 deficiency syndrome are colored in red. N-domain SP-A network in the inward conformation represented by the GLUT1 structure (left) and the outward conformation represented by the GLUT3 structure (PDB 4ZW9) (right). Selected residues are shown as sticks and hydrogen bonds are represented by yellow dashes (2.6–3.6 Å distances). **(B)** GLUT1 E209Q: $K_m$ = 17 ± 3.4 mM and $V_{max}$ = 6,473 ± 528 pmol/oocyte/30 min. **(C)** GLUT3 E207Q: $K_m$ = 6.2 ± 1.2 mM and $V_{max}$ = 1,418 ± 88 pmol/oocyte/30 min. Data information: In (B, C) Michaelis–Menten analysis of 2-DG uptake in oocytes. Data represents the mean ± SD of three or more replicate experiments.

density matches a glucose, but at a lower density threshold additional density appears that fits an aliphatic tail, so in line with our central binding site modeling, we modeled this as an NG molecule (Figs 3A and S4). Although the observed density could be an experimental artifact, in a possible physiological context, this could also reflect lipid headgroup and/or a substrate molecule binding to a potential allosteric binding site because the interaction is to the SP motif, a highly conserved region of SP proteins. Indeed, the D-glucopyranoside headgroup in this "secondary site" establishes polar interactions with Arg93, Asn94, Glu209, Arg218, and Arg223 from both the N-domain SP motif, ICH1-2 and M3 residues (Fig 3A). The headgroup thus supports the charged network of the conserved

cytosolic residues, which in turn moves the SP motif residues away from the A motif residues and stabilizes the inward-open conformation of GLUT1. We note that another structure of GLUT3 (PDB 5C65) contains two octyl glucose neopentyl glycol molecules near this site, but they do not interact with the conserved residues of the SP motif, and their position are more embedded towards the bilayer center suggesting a default lipid/detergent interaction and not D-glucopyranoside headgroup coordination, as in our data. To help evaluate whether the secondary site density was derived from our specific experimental setup, we analyzed the previously published 3.2 Å GLUT1 dataset (Deng et al, 2014), where we also observe a clear density in the map at this site, supporting that a glucose molecule

interacts with the SP motif in the N-domain under different experimental conditions.

## GLUT1 and GLUT3 kinetics can be exchanged with a single-point mutation in the C-domain SP motif

Given the disparities between GLUT1 and GLUT3 kinetics, we sought to identify a possible cause using our high-resolution GLUT1 structure, compared with GLUT3, with the SP-A network in mind. The N-domain SP-A interactions are identical between GLUT1 and GLUT3, so we decided to investigate the homologous SP-A network in the C-domain.

The A and SP motifs are almost identical between the N- and C-domains of GLUT1 and GLUT3 (RMSD(CA) < 0.8 Å), with a notable difference being that Lys456 of GLUT1 is replaced with Arg454 in GLUT3 (Figs 5A and S9). Unlike the SP motif in the N-domain, part of the C-domain SP motif is disordered. In the high-resolution outward facing GLUT3 structure, Arg454 is part of the SP-A network together with Glu452 (Fig 5A). However, the outward-facing conformation of another GLUT3 structure (PDB ID 5C65, 2.65 Å) has the Arg454 pointing out into the solvent, suggesting that binding of Arg454 to the A motif, although clearly possible, might not be a key aspect of a C-terminal SP-A network. The Lys456 residue is not resolved in the GLUT1 structures, but in the inward conformation

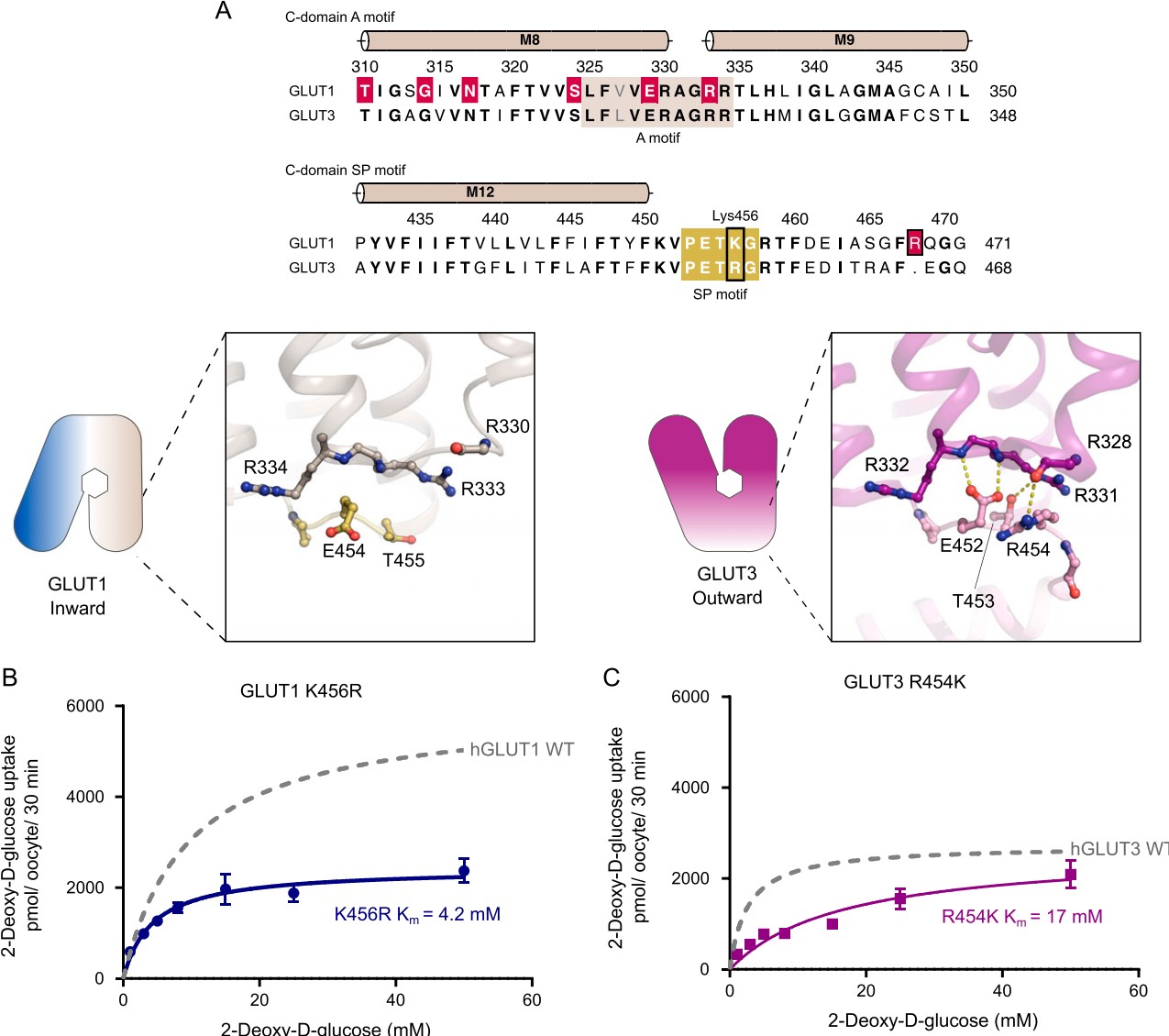

**Figure 5. The C-domain Sugar Porter (SP)-A network between inward and outward states.**
**(A)** Sequence alignment between GLUT1 and GLUT3 of the C-domain A motif (brown) and the C-domain SP motif (yellow). Residues involved in GLUT1 deficiency syndrome are colored in red. C-domain SP-A network in the inward conformation represented by the GLUT1 structure (left) and the outward conformation represented by the GLUT3 structure (PDB 4ZW9) (right). Selected residues are shown as sticks and hydrogen bonds are represented by yellow dashes (2.6–3.6 Å distances). **(B)** GLUT1 K456R: $K_m$ = 4.2 ± 0.6 mM and $V_{max}$ = 2,427 ± 90 pmol/oocyte/30 min. **(C)** GLUT3 R454K: $K_m$ = 17 ± 3.3 mM and $V_{max}$ = 2,638 ± 234 pmol/oocyte/30 min. Data information: In (B, C) Michaelis–Menten analysis of 2-DG uptake in oocytes. Data represent the mean ± SD of three or more replicate experiments.

of the GLUT1 bacterial homolog XylE, the equivalent residue is establishing a salt bridge with an acidic residue from the A motif (Fig S10).

Based on the difference between GLUT1 and GLUT3 SP-motifs, we made the mild mutant GLUT1 K456R, which we speculated would improve substrate affinity by mimicking the GLUT3 isoform perhaps by destabilizing the inward conformation via SP-A network favoritism. We tested this mutant and in support of our model, the $K_m$ values increased by twofold higher apparent affinity (4.2 mM), and there was a 2.5-fold decrease in $V_{max}$, very similar to that of native GLUT3 (Fig 5B). To further support the hypothesis, the invert mutation, GLUT3 R454K was created. The transport affinity of the mutant was strongly affected with a sixfold lower apparent affinity than native GLUT3 ($K_m$ 17 mM) (Fig 5C), again supporting the model. Mutant protein expression levels were found to be equal or higher than the WT protein for all the C-terminal mutants (Fig S8).

We speculate that the arginine in GLUT3, but not the lysine in GLUT1, is able to bind to the A motif in the outward conformation, leading to a stabilization of the outward conformation for GLUT3 compared with GLUT1. This is supported by the high-resolution GLUT3 structure solved in a lipid bilayer, but the alternative GLUT3 structure solved in a detergent environment show that Arg454 is not necessary for stabilization of the GLUT3 outward conformation, and future alternative models are possible.

## Discussion

We have shown that in an identical experimental setup, GLUT3 has a threefold higher transport apparent affinity and a lower turnover than GLUT1, but has similar substrate selectivity. The 2.4 Å resolution structure of GLUT1 allows us to identify several features, including a glucoside-detergent, NG, found in the central sugar-binding site, several water molecules and a PEG molecule bound to a promiscuous binding-pocket suggested to be the binding site of endofacial inhibitors (Kapoor et al, 2016). It also reveals two previously undescribed features in the N-domain: (1) A chloride-ion binding site, unique for the inward conformation, between the SP and A motifs. (2) An intracellular glucose and/or lipid-binding site at the SP motif with a NG molecule bound.

We describe a stabilizing SP-A network found in outward conformations in both N- and C-domain in GLUTs, but also more generally in SP proteins. We show that in the N-domain, disruption of the SP-A network leads to reduced transport affinities, possibly by destabilizing the outward conformation. The SP motif acidic residue has a very subtle, but well-defined, role in generating the SP-A network. We note that GLUT3 residue E207 stabilizes the SP-A network to secure an outward state in a previously published structure (Deng et al, 2015). We have described that at the same position in GLUT1, a chloride ion replaces the acidic group in the inward conformation of the N-domain. When we mutate the glutamate residue of the SP-motif to neutralize the acidic group, uptake assays show a significantly increased $K_m$ value.

It has previously been shown that the A motif strongly influences conformational equilibrium, and that manipulations of the A motif results in kinetic changes in Major Facilitators (Sato & Mueckler, 1999;

Jiang et al, 2013; Zhang et al, 2015). The lipid phosphatidylethanolamine has been suggested to interact with the A motif in other MFS proteins to modulate transport kinetics (Martens et al, 2018). We observe a similar type of interaction here between a glucose or detergent molecule and the SP motif, supporting the hypothetical idea of endofacial regulation by intracellular modulators.

We further analyzed the GLUT1 C-domain SP motif and identified Lys456 as a key residue in kinetic modulation. By introducing the GLUT1 K456R mutation, mimicking GLUT3, GLUT1 kinetics became GLUT3-like. Conversely, the GLUT3 R454K mutation decreased apparent affinity to similar levels as GLUT1. The conservative Arg to Lys modification (and vice-versa) illustrates that a precisely tuned interaction network is crucial for transport activity, as previously seen in Major Facilitator motifs (Jiang et al, 2013).

Based on our results, we suggest a provisional model for kinetic control of the GLUT transport cycle (Fig 6). In this model, the A and SP motifs act as molecular switches which together with intracellular modulators control the activation energy between different conformational states, and in particular the rate-limiting step from empty inward to empty outward conformation. It is likely that the N- and C-domain SP-A networks have slightly different properties. We believe the N-domain SP motif glutamate residue has the most prominent role, as this residue forms hydrogen bonds with the A motif backbone residues, creating the SP-A network in the outward conformation to stabilize it. During the transition to the inward conformation, the SP-A network is broken by the glutamate flipping away from the A motif. In this transition, a chloride ion replaces the glutamate residue, whereas an intracellular modulator neutralizes the charged network of the cytosolic residues in the N-domain. This modulator could be a lipid or glucose in a physiological context. This helps to stabilize the inward conformation, and thus, these intracellular modulators indirectly reduce substrate affinity by keeping the transporter in the inward conformation.

While we were able to identify intracellular modulators that appear to interact with the N-domain SP motif in the inward conformation, no such modulators could be identified in the C-domain. Although we do not exclude the possibility of intracellular modulators playing a role here too, it is worth noting that the GLUT1 M9 transmembrane helix next to the C-domain A motif promotes oligomerization (De Zutter et al, 2013). It is possible that GLUT oligomerization could influence the kinetics of transport between monomers in a higher oligomeric state through the SP-A network.

Our findings suggest that a physiological function of the SP motif is to modulate the kinetics of transport by interacting with the A motif, with an additional function as a structural element that could respond to intracellular modulators such as glucose and lipids. An intermittent use of chloride during the transport cycle would allow for further fine-tuning of this conformationally sensitive network. The suggested role of the SP motif might expand beyond the SP family. We have analyzed the sequences of all known MFS proteins listed in the Transporter Classification Database (TCDB) (Saier et al, 2014) and of 99 MFS families currently found in the database (including the SP family), we have identified four additional MFS families that also have a SP sequence motif similar to the SP family: The Phosphate: H+ Symporter (PHS) family, the Organic Cation Transporter family, the Vesicular Neurotransmitter Transporter family, and

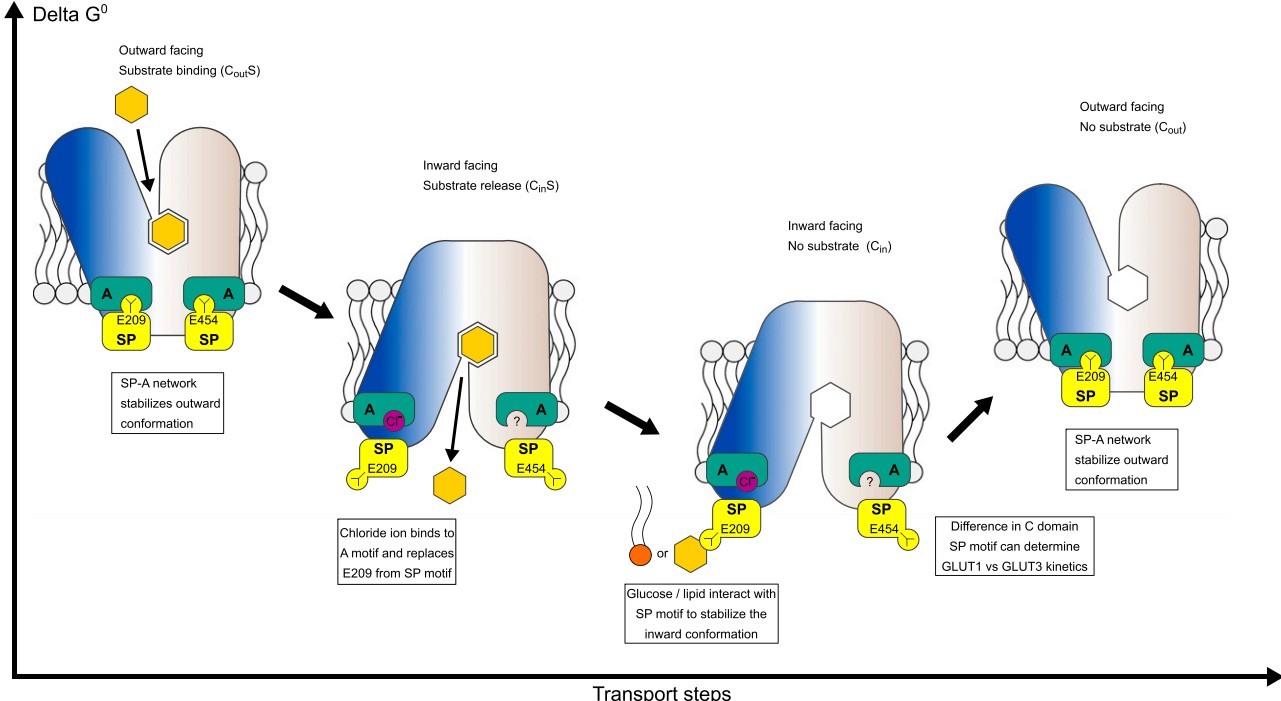

**Figure 6. Model for kinetic control of the transport cycle based on the Sugar Porter (SP)-A network.**
The SP-A network stabilizes the outward conformation where glucose binds from the extracellular side to the central substrate binding site. Glucose binding leads to closure of the binding site towards the extracellular side and the SP motif glutamate flips away from the A-motif, interchanged by a chloride ion. The disruption of the SP-A network opens the central cavity to the intracellular side and a cytosolic exit pathway for glucose is created. Direct interactions between glucose/lipids with the SP and A motifs can stabilize the inward conformation.

the Plant Copper Uptake Porter (Pl-Cu-UP) family. We predict that these families are closely related to the SP family and share elements of the same regulatory mechanism.

In conclusion, we have identified molecular switches that modulate glucose transport in GLUTs, and can explain affinity differences between GLUT1 and GLUT3. We have presented a tentative model that rationalizes how these switches work, which also provides a mechanistic function for the defining SP motif in the SP family. We suggest that the SP motif is a prime candidate for future work to probe and modify transport kinetics in both human GLUT transporters and other SP family members needed for metabolic uptake of sugars in all organisms.

# Materials and Methods

### Protein expression and membrane isolation

For protein expression in *Saccharomyces cerevisiae*, the *GLUT1* gene (UniProt P11166) with a C-terminal deca-histidine tag and a thrombin cleavage site was cloned into an expression plasmid, based on p423_GAL1 (Mumberg et al, 1994; Lyons et al, 2016). Sequence-verified clones were transformed into chemically competent *S. cerevisiae* cells (DSY-5 strain, Cat. no. P04003; vendor GENTAUR) and plated onto a SD/-His synthetic medium plate and incubated at 30°C, for 3 d. Freshly grown colonies were inoculated in 10 ml of synthetic medium without histidine, SD-His (sunrise) and incubated with shaking at

30°C. The following day, the 10 ml culture was inoculated into 1 liter of SD/-His synthetic medium, grown overnight, and then used in a culture vessel to grow to high density. After harvest, the yeast pellet was re-suspended in lysis buffer (100 mM Tris, pH 7.5, 600 mM NaCl, 1.2 mM PMSF) and lysed by running 5 × 1-min cycles on a bead-beating disrupter with 0.5-mm glass beads. The cell lysate was centrifuged for 20 min at 20,000$g$, followed by isolation of membranes by ultra-centrifugation at 200,000$g$ for 150 min. The membranes were homogenized and stored at −80°C.

### Protein purification

Frozen membrane aliquots were thawed on ice and re-suspended in solubilization-buffer (50 mM Tris–HCl, pH 7.5, 250 mM NaCl, and 5% glycerol) with 1% (wt/vol) n-dodecyl-$\beta$-d-maltoside (DDM) and 0.1% (wt/vol) cholesteryl hemisuccinate (CHS). After incubation for 30 min at 4°C, insoluble material was removed by filtration using a 1.2 $\mu$m filter. Imidazole, to a final concentration of 20 mM, was added to the supernatant which was loaded onto a 5 ml Ni-NTA column (GE Healthcare). After loading, the column was washed with 10 column volumes (CVs) of W60 buffer (solubilization buffer with 0.1% [wt/vol] DDM, 0.01% [wt/vol] CHS, and 60 mM imidazole, pH 7.5), followed by 10 CV G-buffer (20 mM MES, pH 6.5, 250 mM NaCl, 10% [vol/vol] glycerol, 0.2% [wt/vol] NG, 0.02% [wt/vol] CHS, 0.5 mM Tris [2-carboxyethyl] phosphine [TCEP], and 40 mM D-glucose). Protein was eluted by circulating G-buffer containing bovine thrombin and 20 mM imidazole at 19°C for ~16 h, followed by a three CV wash with

G-buffer supplemented with 40 mM glucose. The sample was concentrated using a 20 ml concentrator with 100-kD cutoff (Amicon) and further purified by size exclusion chromatography on an Enrich 650 10/300 column (Bio-Rad) in G-buffer. The composition of the G buffer was optimized through a thermostability assay (Tomasiak et al, 2014). Selected peak fractions were pooled and directly used for crystallization trials at ~5 mg/ml. Protein purity was followed through all steps of the purification via SDS gel electrophoresis with InstantBlue Coomassie staining (Expedeon).

### Crystallization

Crystals were grown at 17°C by vapor diffusion in 0.6 + 0.6 μl hanging drops with a reservoir containing 42–46% polyethylene glycol 400, 100–200 mM MgCl$_2$ and 100 mM Mops, pH 7.4. Cubic crystals, with a final size around 100 × 100 × 100 μm, were obtained after 1–3 d of crystal growth.

For anomalous experiments to confirm the Cl$^-$ site, crystals were grown using an identical protocol, but exchanging the reservoir solution MgCl$_2$ to MgBr$_2$, before mixing the 0.6 + 0.6 μl drop.

### Data processing

Data were collected at the Diamond Light Source beamlines i24 and i04, processed and scaled in space group C2 using XDS (Kabsch, 2010). Complete datasets were obtained from single crystals, but an improved dataset was obtained by averaging data from two separate datasets collected sequentially from a single large crystal. The structures were solved by molecular replacement in PHASER (McCoy et al, 2007) using PDB model 4PYP (Deng et al, 2014). Rfree flags were transferred from 4PYP and extended to the resolution of the dataset. COOT (Emsley et al, 2010) and Namdinator (Kidmose et al, 2019) were used for model building, and refinement was performed in phenix.refine (Adams et al, 2010) using a refinement strategy of individual sites, individual ADP and group TLS (three groups) against a maximum likelihood target using reflections in the 42–2.4 Å range. This yielded an R(work) of 20.4% and R(free) of 22.9%. Residues 1–8 and 456–492 were not visible in density maps and were omitted from the final model. MolProbity (Chen et al, 2010) was used for structure validation and gave a Ramachandran plot with 97.1% of residues in favored regions and 0.2% residues in disallowed regions. The rama-Z score for the model was −2.17 (Sobolev et al, 2020).

Crystals grown with MgBr$_2$ diffracted to 3.2 Å. Data were collected at the wavelength near the bromine K-absorption edge (0.9184 Å) to maximize the anomalous signal, and after processing a dataset to 4.0 Å in XDS, the anomalous difference Fourier map was calculated using FFT (Winn et al, 2011). The map showed one single strong anomalous peak (7.3 sigma) confirming the Cl$^-$ site. Structural figures were prepared using PyMOL 1.5.0.4 (The PyMOL Molecular Graphics System [Schrödinger LLC, 2012]). Sequence alignments were constructed with PROMALS3D (Pei et al, 2008), followed by manually refining gaps based on the observed structure. Alignments were visualized using ALINE (Bond & Schüttelkopf, 2009). SP motifs in the MFS were identified using MEME (Bailey et al, 2006) based on all MFS sequences from TCDB, excluding the SP family (883 sequences total in 98 MFS families, not including SP family sequences) (Saier et al, 2014).

### Oocyte transport assays

Human GLUT1 and GLUT3 (UniProt P11169) were cloned into the pNB1-U vector (Nour-Eldin et al, 2006). All mutations were created using the quick-change lightning site directed mutagenesis kit according to the manufacturer's instructions (Agilent Technologies). The GLUT3 chimera was generated by amplifying the C-terminal residues 450–492 of GLUT1 and clone them after residue 447 of GLUT3.

For cRNA preparation, the genes cloned in the pNB1-U vector were first amplified by PCR using standard primers containing the 5' and 3' UTRs: Fw (TTAACCCTCACTAAAGGGTTGTAATACGACTCACTATAGGG) and Rv (TTTTTTTTTTTTTTTTTTTTTTTTTTTTTTTATACTCAAGCTAGCCTCGAG), and then analyzed and purified on a 1% agarose gel. RNA was synthesized using the mMESSAGE mMACHINE T7 Transcription Kit (Thermo Fisher Scientific). Needles for RNA injection were made using the PC-10 Micropipette Puller Narishige. Oocytes from *Xenopus laevis* were purchased from EcoCyte Bioscience (Castrop-Rauxel). For expression, ~25 ng of RNA was injected into oocytes, using the Nanoject III (Drummond Scientific). Before uptake assays, oocytes were incubated at 16°C for 3 d in ND-96 buffer (82.5 mM NaCl, 2 mM KCl, 1 mM MgCl$_2$, 1 mM CaCl$_2$, 5 mM Hepes, pH 7.5) with 10 IU/ml streptomycin (Invitrogen). Uptake assays were performed using concentrations of 2-deoxy-D-glucose (Sigma-Aldrich) ranging from 0 to 50 mM in ND-96 buffer. Radioactively labeled deoxyglucose ([3H]-2-DG) was added to each sugar solution at a concentration of 5 μCi/ml (PerkinElmer). In concentration-dependent assays, each group of five oocytes was incubated with the reaction buffer for 30 min. The reaction was stopped by adding ice-cold ND-96 buffer supplemented with 100 μM Phloretin. For control assays, oocytes were incubated with 20 μM of cytochalasin B in ND-96 buffer for 5 min before the uptake assay. In competition assays, the reaction buffer was composed of 5 μCi/ml [3H]-2-DG, 5 mM deoxy-glucose and 20× fold higher concentration of the competing sugar. Each oocyte was treated as a single experiment. Oocytes were transferred individually to a scintillation vial and disrupted by the addition of 100 μl of a 10% SDS solution followed by immediate vortexing. 3 ml of EcoScintTM H scintillation fluid (National Diagnostics) was added to each sample and radioactivity was quantified using a Tri-Carb 5110TR Liquid Scintillation Counter (PerkinElmer). Before analysis, the measured sugar uptake was corrected for unspecific uptake as follows: Water-injected oocytes were exposed to the same 2-DG concentration and time as the GLUT- injected oocytes. This uptake was subtracted before calculating uptake. Experiments were performed at least in triplicates and data were analyzed with GraphPad Prism 7. Michaelis–Menten fitting was used for curve-fitting analysis and kinetic parameters determination. Error bars represent the SD.

### Quantification of protein in oocyte membranes by Western blot

Preparation of total and plasma membrane sections from oocytes was achieved as described previously (Leduc-Nadeau et al, 2007). For total membranes, five oocytes were rinsed in ND-96 buffer and homogenized in 1 ml of the same solution, supplemented with 0.5–1 mM PMSF, by hand with a P200 pipettor. The homogeneous solutions were centrifuged at 16,000$g$ for 20 min at 4°C to pellet down total membranes. The membrane pellets were resuspended in 10 μl ND-96 buffer with 1% (vol/vol) Triton X-100 and frozen until use. For plasma membranes, oocytes were incubated for 10 min in MBS buffer supplemented with 0.005% subtilisin A (Sigma-Aldrich) under very

mild agitation to partially digest the vitelline membranes. The oocytes were then polymerized at 4°C with 1% ludox and with 0.1% polyacrylic acid (Sigma-Aldrich). The oocytes were homogenized by hand with a P200 pipettor in an Eppendorf tube with 0.5 ml of cold HbA (5 mM $MgCl_2$, 5 mM $NaH_2PO_4$, 1 mM EDTA, 80 mM sucrose, and 20 mM Tris, pH 7.4). The homogenates were diluted to a total volume of 1.5 ml with HbA and centrifuged at 16$g$ for 30 s at 4°C. This last step was repeated three times. Finally, to pellet the purified plasma membranes, the samples were centrifuged at 16,000$g$ for 20 min.

Before Western blotting, proteins from the membrane preparations were separated according to their size by SDS–PAGE gel. Proteins were transfer to a Polyvinylidene fluoride (PVDF) membrane (Millipore), using a semi-dry system (Bio-Rad). First the PVDF membrane was incubated for 60 s with 100% of MeOH. The filter paper, membrane and gel were then incubated for 10 min in transfer buffer (25 mM Tris–HCL, pH 8.3, 150 mM glycine, 10% [vol/vol] methanol). Blotting was performed at room temperature at 18 V for 30 min. The PVDF membrane was blocked in 5% skimmed milk (wt/vol) in TBS-T (10 mM Tris–HCL, pH 8.0, 150 mM NaCl, and 0.05% [wt/vol] Tween-20) for 1 h at room temperature, with rotation. The membrane was incubated with the primary antibody (anti-GLUT1 Monoclonal antibody, 1:5,000) (Abcam) overnight with rotation. The following day, the membrane was washed with TBS-T 3× for 10 min, before secondary antibody incubation ($\alpha$-mouse HRP, 1:1,000) (Sigma-Aldrich). The secondary antibody was incubated for 1 h at room temperature, with rotation. The wash was repeated two times and bound antibody was detected using a chemiluminescent substrate solution (Thermo Fisher Scientific).

## Data Availability

Coordinates and structure factors for GLUT1 have been deposited in the Protein Data Bank with the accession number 6THA.

## Supplementary Information

## Acknowledgements

The authors acknowledge beamlines I24, I04-1, and I04 at the Diamond Light Source, where X-ray data were collected, as well as Max IV Laboratory, DESY-PETRA III, and the Swiss Light Source for crystal screening. We also thank H Nour-Eldin for the pNB1u plasmid backbone used for cloning and oocyte assays. This work was supported by funding from the European Research Council (grant agreement No. 637372), the Novo Nordisk Foundation (grant No. NNF17OC0026900), the Carlsberg Foundation (CF17-0180), an Aarhus Institute of Advanced Studies (AIAS) fellowship to BP Pedersen and the Jeppe Juhl and wife Ovita Juhls Memorial Fund to PA Paulsen.

### Author Contributions

TF Custódio: conceptualization, data curation, formal analysis, validation, investigation, visualization, methodology, and writing—original draft, review, and editing.

PA Paulsen: conceptualization, data curation, formal analysis, validation, investigation, visualization, methodology, and writing—original draft, review, and editing.

KM Frain: investigation, visualization, methodology, and writing—original draft, review, and editing.

BJ Pedersen: conceptualization, supervision, funding acquisition, validation, investigation, visualization, project administration, and writing—original draft, review, and editing.

### Conflict of Interest Statement

The authors declare that they have no conflict of interest.

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
