## [Reviewer comments · Life Science Alliance]

Life Science Alliance

Structural comparison of GLUT1 to GLUT3 reveal transport regulation mechanism in Sugar Porter family

Tânia Custódio, Peter Paulsen, Kelly Frain, and Bjørn Pedersen

DOI: <https://doi.org/10.26508/lsa.202000858>

Corresponding author(s): Bjørn Pedersen, Aarhus University

Review Timeline:

Submission Date:	2020-07-23
Editorial Decision:	2020-08-26
Appeal Received:	2020-08-31
Editorial Decision:	2020-09-16
Revision Received:	2020-12-19
Editorial Decision:	2021-01-12
Revision Received:	2021-01-18
Accepted:	2021-01-19

Scientific Editor: Shachi Bhatt

Transaction Report:

August 26, 2020

Re: Life Science Alliance manuscript #LSA-2020-00858

Dr. Bjørn Panyella Pedersen
Aarhus University
Department of Molecular Biology and Genetics
Gustav Wieds Vej 10
MBG-AU
Aarhus C, Danmark 8000
Denmark

Dear Dr. Pedersen,

Thank you for submitting your manuscript entitled "Structural comparative analysis of GLUT1 to GLUT3 and uptake regulation in the Sugar Porter family" to Life Science Alliance (LSA). Your manuscript has been reviewed by the editors and outside referees (referee comments below). As you will see from the reports below, while the manuscript is interesting for LSA and for the referees, the reviewers are concerned that the conclusion - presence of a chloride binding site in GLUT1 is not fully supported by findings presented, and would require experimental evidence.

Given the interest in the topic, we would be open to resubmission to LSA of a significantly revised and extended manuscript that fully addresses the reviewers' concerns, particularly includes experimental evidence in support of chloride binding site in GLUT1, is significantly rewritten to correct for grammatical errors, and is subject to further peer-review. If you would like to resubmit this work to LSA, please submit an appeal directly through our manuscript submission system with a revised manuscript. Please note that priority and novelty would be reassessed at resubmission.

If you wish to expedite publication of the current data, it may be best to pursue publication at another journal.

Regardless of how you choose to proceed, we hope that the comments below will prove constructive as your work progresses. We would be happy to discuss the reviewer comments further once you've had a chance to consider the points raised in this letter.

Thank you for thinking of Life Science Alliance as an appropriate place to publish your work.

Sincerely,

Shachi Bhatt, Ph.D.
Executive Editor
Life Science Alliance

Reviewer #1 (Comments to the Authors (Required)):

The authors explore the role of two conserved regions in the sugar porter family (SP), the SP and A motifs, on the substrate affinity and V_{max} of transport in two glucose transporters GLUT1 and GLUT3, which share identical binding sites for the substrate. Their crystal structure of the wild-type GLUT1 in the inward-facing conformation shows 3 new ligands: an NG molecule close to the SP motif, a Cl^- ion at the interface between the SP and A motifs, and a PEG molecule in the binding site for GLUT1 inhibitors. The authors explore the interactions between the SP and A motifs through site-directed mutagenesis and functional studies, finding that a very minor substitution in the C-half SP motif Lys/Arg can interconvert kinetically GLUT1 and GLUT3. They propose a model in which the transport of GLUTs can be modulated by intracellular effectors (ions, lipids or sugars) through their action on the SP or SP-A network to hasten or delay the inward- to outward-facing transition. This report tackles the role of the soluble loops on the mechanism of transport of GLUTs, providing evidence that differences in substrate affinity are about more than the active site in SP members and, more generally, MFS transporters. Concerns arise from the overinterpretation of the structural data, particularly the roles for the Cl^- site and the secondary NG site, which could be simply crystallization artefacts due to crystallization and protein purification conditions.

- Major concerns

1. Given the high concentration of Cl^- used in crystallization (100-200 mM) the Cl^- site may be a crystallization artefact and not a biologically relevant site. Also, it's unclear from the paper how Cl^- is recognized through interactions with the protein residues (show distances to relevant atoms in the figure 3) to support the notion of a specific ion site. Moreover, when looking at the C-domain A and SP motives (Fig. 4B), E454 also has its side-chain away from the backbone amides of R334 and R333, without a Cl^- forcing it away. Therefore, this conformation of E209/E454 side-chain occurs irrespective of the competition with a Cl^- , the binding of Cl^- is probably opportunistic. Additional experiments (for example ITC) that show Cl^- binding to GLUT1 are necessary to support the authors model (Fig. 6).

2. The secondary site of NG is similar to that in the outward-facing conformation of GLUT3 (PDB ID 5C65) for LMNG. The paragraph about this site not being present in the outward-facing conformation of GLUT3 (page 5) needs to be amended. In GLUT3 (PDB ID 5C65), at this site, two detergent molecules sandwich the side-chain of F204 with their hydrophobic tails, while their glucosyl moieties interact with the side-chains of R91 and R228 and the carbonyl of P203, suggesting opportunistic detergent-protein interaction. Thus, interactions at this site seem more related to the protein interactions with surrounding lipids rather than a new effector site.

3. R223 mutants of GLUT1 are very difficult to understand. R223N and R223P have almost no effect, but R223Q decreases the substrate affinity by almost 3-fold! Maybe double-check the sequence for the mutant and redo the kinetic experiment for K_m determination.

4. On page 7, the hypothesis that R454 of GLUT3 vs. K of GLUT1 leads to differences in the outward-facing conformation stabilization due to R being able to bind to the A motif while K can not is not supported by the outward-facing conformation GLUT3 structure (PDB ID 5C65) in which R454 does not interact with the A motif. Also, the interaction for R454 side-chain with the carbonyl of R328 (Fig. 5C) is one H-bond which should be easily accomplished by a Lys residue as well. Therefore, this explanation/hypothesis does not hold up. Maybe dynamic simulations of the transporter embedded in lipids, having either R or K in the position of R454 would help in better understanding the swap on GLUT1 and GLUT3 kinetics.

- Minor concerns

1. There are a lot of grammar mistakes.

2. How was the protein amount calculated from the Western to be able to estimate the V_{max} (for ex., lines 103-104 on pg. 3)? How do you know that the protein level of GLUT3 (wild-type) and GLUT3-chimera (designed to be recognized by the same GLUT1 Ab) are the same?

3. It would help to specify in the text the SP and A motifs in terms of the GLUT1 sequence, besides having them in the figure (for ex. on page 5).

4. Incorrect reference for the inward-facing conformation of XyleE in the first line on page 7.

5. Page 7, lines 235-6, it should be K_m value "decreased" instead of "increased" to 6 mM.

6. Why GLUT1 and GLUT3 molecular weights are very different in the Fig. S8?

7. Fig. S3A needs to show a Western Blot.

8. Several scientists communicated with me that inhibitors in PDB 5EQG and 5EQH do not inhibit GLUT1 (I did not confirm it in my lab). I am not even sure the electron density of cytochalasin B. Kapoor et al. might see only PEG.

Reviewer #2 (Comments to the Authors (Required)):

This paper on the basis of mutation studies and comparison of differences between their effects on GLUT1 and GLUT3 kinetic parameters K_m and V_m on glucose uptake into transporters expressed in oocytes suggests that the A and SPA sites at the endofacial surface have affinities with lipid and Cl^- . In the case of GLUT1 Cl^- impedes this interaction and hence is responsible for its slightly lower affinity for glucose than GLUT3.

The paper has not been particularly well prepared: Two errors in the reference list - Cain is mentioned in the text - but not in the reference list and Viitanen appears to be a co-author with Kaback - regrettably but not of the paper in question Structure of YajR as suggested in the bibliography!.

These minor problems apart - a major defect is that the main new finding upon which the authors place considerable emphasis - namely the Cl^- binding site Fig 3. Although mutations R223P and R223Q show deviations from WT kinetics may reflect changes in the affinity of Cl^- but I do not see any statistical significance reported with regard to differences to WT kinetics. The errors of the parameters K_m and V_m are reported but no estimates of their significant deviations from WT values is reported. This is a pity as obviously it is important to know whether the mutations produce real changes in these parameters or whether we just have to take on board the authors' assertions that they are.

The obvious kinetic experiment Cl^- replacement has not been done or at least not reported. Literature search has revealed a paper - Bissonnette Jm et al Journal of Membrane Biology 58 75-80 Glucose uptake into plasma membrane vesicles from the maternal surface of human placenta "Uptake of D-glucose exceeded that of L-glucose. The uptake of D-glucose was not enhanced by

placing 100 mM NaCl or NaSCN in the medium outside the vesicles (none inside) at the onset of uptake determinations. D-glucose transport was inhibited by cytochalasin B; phloretin, phlorizin, and 1-fluoro-2,4-dinitrobenzene". Not exactly a definitive paper but nevertheless suggests that Cl replacement is without much obvious effect on glucose uptake.... GLUT1 is expressed in human placenta "Localization of erythrocyte/HepG2-type glucose transporter (GLUT1) in human placental villi " K Takata, T Kasahara, M Kasahara, O Ezaki... - Cell and tissue ..., 1992.

These are rather negative comments and I do not wish to be too discouraging. A role for ligand binding to these endofacial linker motifs is an interesting possibility and is obviously worth more thorough investigation not least with molecular dynamics.

However, the manuscript needs to be critically reviewed and more sharply focussed. A more critical consideration should be applied to a discussion as to whether and how ligand binding at the endofacial surface can really alter the K_m for net glucose import as claimed apart from the disputed thermodynamic and kinetic arguments that is.

Other minor points - are the claimed ligand binding sites for PEG and nonyl glucose relevant ? or simply junk binding due to the crystallographic preparation methods? Are these sites present in GLUT3?

In summary the obvious novelty in this paper is the chloride binding site in GLUT1 but the mutation studies are at best loosely supportive of any functional role for Cl⁻ at this site. More direct experimental work should be done with anion replacements to support or refute the significance of this finding.

Reviewer #3 (Comments to the Authors (Required)):

Tania et al report a crystal structure of a human glucose transporter in the inward-open and substrate-bound form. Careful analysis of crystallography at 2.4Å resolution with sufficient statistics as well as oocyte assay determine a cytoplasmic chloride binding site and a candidate regulatory site. Comparison between closely related neuron GLUT3 reveals a remarkable role of SP motif in transport regulation, and explains at least in part, the difference of the apparent glucose affinities between GLUT1 and GLUT3 by regulating their conformational equilibrium. Given these advances, this reviewer agrees that this paper will be of specific interest to the field of membrane transporters and of general interest to the much larger fields of membrane transport mechanisms.

In the manuscript, effect of SP-A interaction is carefully investigated by either structural comparison with other GLUT transporters, and by the mutagenesis studies on two isoforms GLUT1 and GLUT3. However, it is unclear how SP-A network allosterically modulate the conformation of whole molecule. Regulatory sites (SP-A) in the N and C domain seems far from the canonical transport site which is located at the interface between two lobes. It is helpful for readers if authors describe how the changes occurred in the SP-A site transmit and affect to the whole transporter structure.

The term "affinity" should be defined correctly. Determined "affinity" from the Michaelis-Menten fitting of the glucose transport is "apparent affinity", as this measurement does not determine the direct binding of glucose to the transporter.

Careful proof reading is recommended, especially about the description of K_m and V_{max} . " K " and " V " should be in *Italic*, and " m " and " max " should be in subscript.

L219

A motif

Fig2D

It is better for reader to change the color of C-domain SP motif to discriminate C-domain A motif.

August 31, 2020

Dear Reilly Lorenz,

The authors of manuscript #LSA-2020-00858 have requested an appeal. Their comments are below.

Dear Dr. Bhatt,

Thanks for the comments and reviews on our manuscript.

Naturally we were a bit disappointed with the conclusion, and I believe that the reviewers, while have several very insightful and good suggestions for improvements, also misunderstood a few things.

In particular, we agree with the reviewers that the major new finding in the paper is the identification of a Cl⁻ ion site, not previously described in the GLUT family (or anywhere else in the entire Major Facilitator Family). Based on the reviewers comments further experimental evidence for this site was suggested in the editorial decision. Unfortunately this is not as easy as it might seem:

1) Anomalous scattering from X-ray crystallography is the de-facto gold-standard for identifying new ion sites in protein structures. We provide very strong evidence for the site as observed in the data. In humans the intracellular Cl⁻ concentration range from 4-100 mM (depending on cell type, and measurement method). We agree that this is below the concentration we used (100-200 mM), but it is not significantly lower, and there are a plethora of papers in the scientific literature where ion sites as the one we describe has been demonstrated on the basis of this type of data.

2) A very relevant question then arises on the physiological importance of such a site. Indeed biochemical experiments to ascribe this would be highly desirable and we have contemplated this for a long period of time. We believe that the mutations we have made are a good foundation for exploring the relevant region of the protein.

Titration of Cl⁻ would offhand be the obvious experiment, as suggested by both reviewer #1 and #2. However there is no clear experimental setup that can address this problem. The in vivo assays we use cannot be done in a Cl⁻ ion depleted fashion. ITC as suggested by reviewer #1 is extremely unlikely to be successful. ITC derives its signal from the 'heat' change in the sample when binding a ligand, and this is directly correlated with binding affinity. Thus ITC is a method that work well with high affinity ligands, but useless if the affinity is in the mM range, which would clearly be the case here. An additional aggravating aspect is that the sample would have to be purified in the absence of Cl⁻ ions which would be an extremely challenging thing to do while maintaining protein integrity.

We have discussed this issue much, before and also again now, and our conclusion has been that the only relevant experiment possible would be a proteoliposome assay. We have been pursuing these for over a year, but obtaining activity has not been successful so far, and if this methods would be successful it would warrant a new manuscript, repeating all of the measurements we have currently performed in oocyte uptake assays.

The other major concerns from reviewer 1 are insightful, but readily addressed by textual changes. Likewise reviewer #2's major concern is focused on the 223 mutants where she/he asks for statistical significance. This is also easy to obtain from the existing data. In addition mutant 223 is not relevant for the Cl⁻ ion site, but the reviewer might have missed this point. Reviewer #3 is positive.

We agree that the language and structure of the paper could benefit from a makeover, and we have engaged a native speaker to help us with this and streamline the manuscript.

In summary, we are happy to provide a new and major revised version of the manuscript for LSA for further peer-review, but additional experimental evidence for the CI- site is not possible at this point. Would such a manuscript be of interest?

I would be happy to discuss the matter further by zoom (#881 133 5521) or phone (+45 2972 3499).

Thanks again for all the input and suggestions for our story.

All the best,

/Bjørn (on behalf of all authors)

--

Bjørn Panyella Pedersen

Associate Professor

Dept. of Molecular Biology and Genetics - Aarhus University

Gustav Wieds Vej 10, DK-8000 Aarhus, Denmark

Phone: +45 2972 3499 | E-mail: bpp@mbg.au.dk | Web: <http://www.pedersenlab.dk>

You can accept or decline this request from the manuscript using the following link:

<https://lsa.msubmit.net/cgi-bin/main.plex?el=A4Na4WQ5A7Chsi1F6A9fd7KKO78fExKNdHLJ1k5Q6AZ>

Sincerely,

Reilly Lorenz

Manuscript Coordinator

Life Science Alliance

September 16, 2020

MS: LSA-2020-00858

Dr. Bjørn Panyella Pedersen
Aarhus University
Department of Molecular Biology and Genetics
Gustav Wieds Vej 10
MBG-AU
Aarhus C, Danmark 8000
Denmark

Dear Dr. Pedersen,

Thank you for contacting us about your manuscript "Structural comparative analysis of GLUT1 to GLUT3 and uptake regulation in the Sugar Porter family" [LSA-2020-00858]. Based upon your response to the concerns raised we would be happy to send a revised paper out to re-review. We understand that you will not be able to provide additional experimental evidence for the Cl- site, but encourage you to address all the other points raised by the referees.

When revising the manuscript for re-review, we suggest that you re-format it as per Life Science Alliance's guidelines (<https://www.life-science-alliance.org/authors>) and provide a point-by-point rebuttal for the reviewers' concerns.

Please use the following link to submit your manuscript:

<https://lsa.msubmit.net/cgi-bin/main.plex?el=A4Na5WQ7A4Cjpd2I1B9ftdnI0oz4QgVdu9ckNVoYkIGgZ>

Yours sincerely,

Shachi Bhatt, Ph.D.
Executive Editor
Life Science Alliance

Reply to reviewers comments [LSA-2020-00858]
(authors reply in blue)

Reviewer #1:

The authors explore the role of two conserved regions in the sugar porter family (SP), the SP and A motifs, on the substrate affinity and V_{max} of transport in two glucose transporters GLUT1 and GLUT3, which share identical binding sites for the substrate. Their crystal structure of the wild-type GLUT1 in the inward-facing conformation shows 3 new ligands: an NG molecule close to the SP motif, a Cl⁻ ion at the interface between the SP and A motifs, and a PEG molecule in the binding site for GLUT1 inhibitors. The authors explore the interactions between the SP and A motifs through site-directed mutagenesis and functional studies, finding that a very minor substitution in the C-half SP motif Lys/Arg can interconvert kinetically GLUT1 and GLUT3. They propose a model in which the transport of GLUTs can be modulated by intracellular effectors (ions, lipids or sugars) through their action on the SP or SP-A network to hasten or delay the inward- to outward-facing transition. This report tackles the role of the soluble loops on the mechanism of transport of GLUTs, providing evidence that differences in substrate affinity are about more than the active site in SP members and, more generally, MFS transporters. Concerns arise from the overinterpretation of the structural data, particularly the roles for the Cl⁻ site and the secondary NG site, which could be simply crystallization artefacts due to crystallization and protein purification conditions.

We have addressed these concerns specifically in the sections below.

- Major concerns

1. Given the high concentration of Cl⁻ used in crystallization (100-200 mM) the Cl⁻ site may be a crystallization artefact and not a biologically relevant site. Also, it's unclear from the paper how Cl⁻ is recognized through interactions with the protein residues (show distances to relevant atoms in the figure 3) to support the notion of a specific ion site. Moreover, when looking at the C-domain A and SP motives (Fig. 4B), E454 also has its side-chain away from the backbone amides of R334 and R333, without a Cl⁻ forcing it away. Therefore, this conformation of E209/E454 side-chain occurs irrespective of the competition with a Cl⁻, the binding of Cl⁻ is probably opportunistic. Additional experiments (for example ITC) that show Cl⁻ binding to GLUT1 are necessary to support the authors model (Fig. 6).

For reference, anomalous scattering from X-ray crystallography is the de-facto gold-standard for identifying new ion sites in protein structures. We thus provide very strong evidence for the chloride site as observed in the data. The relevant question, as stated by the reviewer is whether the site is an artifact or a site with physiological relevance. We believe the chloride binding is not an artifact; human intracellular chloride concentrations range from 4-100 mM (depending on cell type and measurement method), which is only slightly below the concentration used in the experiments (100-200 mM). We have included this line of argumentation in the manuscript, along with more information on how Cl⁻ is recognized through interactions with protein residues to be more specific and allow the reader to draw their own conclusions on the site and its relevance for the described SP-A network. The rewritten paper is significantly more focused on the N domain work (E209) and not the C domain, where we find the clearest conclusion can be drawn. The relevant glutamates are demonstrated to have two distinct conformations, and while the glutamate side-chains can clearly face away from the backbone amides of the arginine's without the chloride present, we believe this is somewhat missing the point: The chloride helps change the energetics of this change when the glutamate is forced away to break the SP-A network in the inward conformation. The chloride is clearly not essential for this dynamic, but the binding helps to create a more stabilized glutamate outward conformation, thus stabilizing the inward conformation in SP proteins. We have rewritten the manuscript to better explain and expand on the key, but subtle, point of the observed chloride site.

A major point in the manuscript is the description of a transient SP-A network that exists in the outward conformation but is broken in the inward conformation, in the process creating a transient chloride site. We propose

that the dynamics of the SP-A network govern the transition between states and thus directly affect protein kinetics. This is the key insight derived from the chloride site observation in the data.

ITC is unfortunately extremely unlikely to be successful. ITC derives its signal from the 'heat' change in the sample when binding a ligand, and this is directly correlated with binding affinity. Thus ITC is a method that is the gold-standard for high affinity ligands, but is impossible to utilize if the affinity is in the mM range, which would clearly be the case here. An additional aggravating aspect is that the sample would have to be purified in the absence of Cl⁻ ions which would be extremely challenging while maintaining protein integrity. In addition we believe the site to be transient, and only exist in the inward conformation, which would further complicate any analysis of a sample purified in a Cl-free state.

A Cl⁻ replacement experiment is non-trivial to perform in a meaningful manner, but could perhaps be done in future proteoliposome assays. We are currently pursuing these but unsuccessfully so far. We believe that proteoliposome assays would be extremely interesting, but also best presented in a full repeat of all the oocyte uptake assays shown here in the manuscript, essentially in a makeover of the entire biochemical analysis. We will reserve such an analysis and makeover for followup studies.

2. The secondary site of NG is similar to that in the outward-facing conformation of GLUT3 (PDB ID 5C65) for LMNG. The paragraph about this site not being present in the outward-facing conformation of GLUT3 (page 5) needs to be amended. In GLUT3 (PDB ID 5C65), at this site, two detergent molecules sandwich the side-chain of F204 with their hydrophobic tails, while their glucosyl moieties interact with the side-chains of R91 and R228 and the carbonyl of P203, suggesting opportunistic detergent-protein interaction. Thus, interactions at this site seem more related to the protein interactions with surrounding lipids rather than a new effector site.

We have amended the paragraph comparing GLUT3 to include 5C65 and its two observed OGN molecules (lines 199-210). While we agree the two detergent molecules are bound by normal and expected detergent-protein interaction in 5C65, we do find that there are some differences to the observation in our data.

The site we describe is much more extended out of the plane of the membrane and the glucose head-group of the modeled NG is clearly engaging in specific interactions with fully conserved residues of the SP motif as described in the manuscript.

We have significantly toned down our conclusion on this point, and highlight that the interaction could be an experimental artifact, but still believe that the observation warrants the reader's attention.

3. R223 mutants of GLUT1 are very difficult to understand. R223N and R223P have almost no effect, but R223Q decreases the substrate affinity by almost 3-fold! Maybe double-check the sequence for the mutant and redo the kinetic experiment for K_m determination.

We agree this information was unclear. These data were an attempt to include an analysis of GLUT1 Deficiency Syndrome and perhaps relate the disease to the observed site, but it was not an attempt to tease out the general intricacies of the chloride site. However, we find that this part had no clear-cut conclusion and detracted from the key message of the manuscript. Therefore we have removed this data in the new focused manuscript.

4. On page 7, the hypothesis that R454 of GLUT3 vs. K of GLUT1 leads to differences in the outward-facing conformation stabilization due to R being able to bind to the A motif while K cannot is not supported by the outward-facing conformation GLUT3 structure (PDB ID 5C65) in which R454 does not interact with the A motif. Also, the interaction for R454 side-chain with the carbonyl of R328 (Fig. 5C) is one H-bond which should be easily accomplished by a Lys residue as well. Therefore, this explanation/hypothesis does not hold up. Maybe dynamic simulations of the transporter embedded in lipids, having either R or K in the position of R454 would help in better understanding the swap on GLUT1 and GLUT3 kinetics.

We are very grateful to the reviewer for pointing out this key point which we had missed in the original manuscript, and we agree our original theory regarding R454 is not well supported by pdb 5C65. We have included this in the new version of the manuscript (lines 241-245).

We also agree using molecular dynamics is an interesting future aspect of this work. Currently we are limited by the lack of the same protein in two distinct conformations (eg. GLUT1 in inside and outside conformation), as well as the limits imposed by the simple bilayer membranes routinely employed in MD. Very likely the effect of the R454K mutation will be highly dependent on lipid composition of the membrane and it lies beyond the scope of this paper to explore this dynamic with current state-of-the-art MD.

- Minor concerns

1. There are a lot of grammar mistakes-

We have rewritten the entire manuscript. In the process we have corrected these mistakes.

2. How was the protein amount calculated from the Western to be able to estimate the V_{max} (for ex., lines 103-104 on pg. 3)? How do you know that the protein level of GLUT3 (wild-type) and GLUT3-chimera (designed to be recognized by the same GLUT1 Ab) are the same?

V_{max} was calculated by measuring radioactive deoxyglucose uptake count and converting it into pmol/oocyte/min. Thus, V_{max} calculations are not based on protein amount, and a comparison of V_{max} values assume that expression levels are comparable between oocytes. Protein expression levels were measured using western blots as an aid to confirm that protein expression levels were comparable (i.e.- mutant and WT expression levels were similar). We have stated this more clearly in the new manuscript. cf. lines 101-102.

We opted for using the same antibody to control expression levels of GLUT1 and GLUT3 to allow samples to be run on the same Western Blot. Therefore there is an underlying assumption is that GLUT3-wt and GLUT3-chimera expression levels are comparable.

3. It would help to specify in the text the SP and A motifs in terms of the GLUT1 sequence, besides having them in the figure 4/5 (for ex. on page 5).

We have added this information in the introduction lines 48 and 54.

4. Incorrect reference for the inward-facing conformation of XylE in the first line on page 7

This has been amended.

5. Page 7, lines 235-6, it should be K_m value "decreased" instead of "increased" to 6 mM-

This has been corrected.

6. Why GLUT1 and GLUT3 molecular weights are very different in the Fig. S8?

This also came as a surprise to us. GLUT1 normally runs like GLUT3 at the size of ~40 kDa. The molecular weights of GLUT1 and GLUT3 are 56 kDa and 52 kDa, respectively, but due to their hydrophobic nature, membrane proteins will generally run faster than their size would predict on SDS-PAGE gels, which is also confirmed in Fig S8B for GLUT3 and Fig S3A for GLUT1.

Both GLUT1 and GLUT3-CHIMERA however run at 55 kDa (e.g. Fig S2). We are investigating this interesting discrepancy. However, for the overall conclusion that the protein expression levels are comparable, this observation is only tangential. As stated above, the Western Blot data is only used to validate that the V_{max} calculations 'per oocyte' are reasonably comparable.

7. Fig. S3A needs to show a Western Blot.

Unfortunately, we do not have a western blot that would match the displayed gel.

We think the SDS-PAGE gel appropriately demonstrates (in S3A) the approximate quality and quantity of GLUT1 as purified by SEC, as needed for the crystallography experiments done in the manuscript. The identity of the samples is conclusively assured by Mass-Spec in earlier preps (data not shown), and by the resultant crystals which yield electron density maps of a quality that would allow discrepancies in sequence to be discovered.

8. Several scientists communicated with me that inhibitors in PDB 5EQG and 5EQH do not inhibit GLUT1 (I did not confirm it in my lab). I am not even sure the electron density of cytochalasin B. Kapoor et al. might see only PEG.

While we did not test GLUT1 inhibitors in this paper, so cannot comment on their efficiency (besides the canonical inhibitor cytochalasin B), we agree the electron density of the inhibitors in 5EQG and 5EQH in Kapoor *et al* might be a PEG molecule. The density found for cytochalasin B does have distinct features that would support modeling cytochalasin B and not PEG. An explanation not discussed in Kapoor et al or Deng et al. (where an unmodeled density is also present at this site) would be that this is a promiscuous site in the crystals that would bind any molecule, either PEG if present or the suggested inhibitors. We have added a sentence in the figure legend of S6 regarding this.

Reviewer #2:

This paper on the basis of mutation studies and comparison of differences between their effects on GLUT1 and GLUT3 kinetic parameters K_m and V_m on glucose uptake into transporters expressed in oocytes suggests that the A and SPA sites at the endofacial surface have affinities with lipid and Cl⁻. In the case of GLUT1 Cl⁻ impedes this interaction and hence is responsible for its slightly lower affinity for glucose than GLUT3.

We thank the reviewer for the useful comments that has clarified a number of problems with how the story was originally presented.

In particular with regards to the above comment: We believe the chloride site is likely present in all SP proteins, also GLUT3, but is a transient site that only exist during the inward facing conformation. We believe this key point is made much clearer in the revised version of the manuscript.

The paper has not been particularly well prepared: Two errors in the reference list - Cain is mentioned in the text- but not in the reference list and Viitanen appears to be a co-author with Kaback - regrettably but not of the paper in question Structure of YajR as suggested in the bibliography!

These reference errors have been amended. We have carefully checked the reference list.

These minor problems apart -a major defect in that the main new finding upon which the authors place considerable emphasis -namely the Cl⁻ binding site Fig 3. Although mutations R223P and R223Q show deviations from WT kinetics may reflect changes in the affinity of Cl⁻ but I do not see any statistical significance reported with regard to differences to WT kinetics. The errors of the parameters K_m and V_m are reported but no estimates of their significant deviations from WT values is reported. This is a pity as obviously it important to know whether the mutations produce real changes in these parameters or whether we just have to take on board the authors' assertions that they are.

Firstly, the data showing changes related to mutations of residue 223 have been removed. These data were an attempt to include an analysis of GLUT1 Deficiency Syndrome and perhaps relate the disease to the observed site, but it was not an attempt to tease out the general intricacies of the chloride site. However, we find that this

part had no clear-cut conclusion and detracted from the key message of the manuscript. Therefore we have removed this data in the new manuscript version.

Secondly, we apologize that there was an omission in figure 1E,F that displays the oocyte competition assays. We have now included a Student's *t*-test for the significance of these data. With respect to a statistical deviation of K_m and V_{max} parameters, we must admit that we are unsure of what exactly the reviewer is proposing. An example from the literature would be very instructive here. We have investigated the literature, but we were not able to identify any examples of the proposed analysis. All studies we have seen (a good example being the very elegant paper from Hresko et al (2016), doi: 10.1074/jbc.M116.730168) do not include the suggested type of analysis and our current stance would there be that the presented statistical information in the current version of the paper is up to present oocyte assay analysis standards. We would be happy to be instructed in how to improve on the current state, as the raw data is readily available.

The obvious kinetic experiment Cl⁻ replacement has not been done or at least not reported. Literature search has revealed a paper- Bissonnette Jm et al Journal of Membrane Biology 58 75-80 Glucose uptake into plasma membrane vesicles from the maternal surface of human placenta: "Uptake of D-glucose exceeded that of L-glucose. The uptake of D-glucose was not enhanced by placing 100 mM NaCl or NaSCN in the medium outside the vesicles (none inside) at the onset of uptake determinations. D-glucose transport was inhibited by cytochalasin B; phloretin, phlorizin, and 1-fluoro-2,4-dinitrobenzene". Not exactly a definitive paper but nevertheless suggests that Cl⁻ replacement is without much obvious effect on glucose uptake.... GLUT1 is expressed in human placenta "Localization of erythrocyte/HepG2-type glucose transporter (GLUT1) in human placental villi " K Takata, T Kasahara, M Kasahara, O Ezaki... - Cell and tissue ..., 1992.

A major point in the manuscript is the description of a transient SP-A network that exists in the outward conformation but is broken in the inward conformation, in the process creating a transient chloride site. We propose that the dynamics of the SP-A network govern the transition between states and thus directly affect kinetics of the protein. We do not believe that the chloride site is regulatory, at least in a traditional sense of the word. We hope that after our edits this is now clear in the manuscript. The site is a more fundamental part of the basic function of SP proteins. Since chloride levels in the cell are relatively stable, regulation by chloride availability would not, as far as we can tell, be a very efficient regulatory mechanism.

A Cl⁻ replacement experiment is non-trivial to perform in a meaningful manner, but could perhaps be done in future proteoliposome assays. We are currently pursuing these but so far unsuccessfully. We believe that proteoliposome assays would be extremely interesting, but would also be best presented in a full repeat of all the oocyte uptake assays show here in the manuscript, essentially in a makeover of the entire biochemical analysis. We will reserve such an analysis and makeover for followup studies.

These are rather negative comments and I do not wish to be too discouraging. A role for ligand binding to these endofacial linker motifs is an interesting possibility and is obviously worth more thorough investigation not least with molecular dynamics.

We agree using molecular dynamics is an interesting future option based on this work. Currently we are limited by the lack of the same protein in two distinct conformations (eg. GLUT1 in inside and outside conformation), as well as the limits imposed by the simple bilayer membranes routinely employed in MD. Very likely the proposed effect will be highly dependent on lipid composition of the membrane and it lies beyond the scope of this paper to explore this dynamic with current state-of-the-art MD.

However, the manuscript needs to be critically reviewed and more sharply focused. A more critical consideration should be applied to a discussion as to whether and how ligand binding at the endofacial surface can really alter the K_m for net glucose import as claimed apart from the disputed thermodynamic and kinetic arguments that is.

We agree that the original manuscript was lacking focus. We have completely rewritten the manuscript and a much sharper focus on the N domain SP motif.

Other minor points - are the claimed ligand binding sites for PEG and nonyl glucose relevant? or simply junk binding due to the crystallographic preparation methods? Are these sites present in GLUT3?

We believe the NG site could possibly reflect a potential regulatory site because NG's D-glucopyranoside headgroup establishes polar interactions with Arg93, Asn94, Glu209, Arg218, and Arg223 from both the N-domain SP motif, ICH1-2 and M3 residues, all highly conserved residues in GLUT sequences and all other SP members. This is discussed in more detail in the revised manuscript. We have added a sentence regarding a previously observed detergent molecule binding to GLUT3 (pdb model 5C65) at a different site. This we believe is an example of opportunistic binding, since it is more buried in the expected membrane interface and only interact with a few non-conserved residues on the protein surface.

We find that our observation of PEG binding is interesting in the context of previously identified inhibitor sites in GLUT1, located at the same position in the protein. We have expanded the discussion of this in the legend of figure S6.

In summary the obvious novelty in this paper is the chloride binding site in GLUT1 but the mutation studies are at best loosely supportive of any functional role for Cl⁻ at this site. More direct experimental work should be done with anion replacements to support or refute the significance of this finding.

We agree a major novelty of the structural work lies in the identified chloride site, and the novel understanding of the SP-A network we identify in this work, and how that relates to conformational change. As discussed in the manuscript and above, the effect of the chloride site is likely very subtle, and the impact lies perhaps less in the chloride site itself and more in the broader understanding of how the SP and A motif can interact to control conformational change.

Reviewer #3:

Tania et al report a crystal structure of a human glucose transporter in the inward-open and substrate-bound form. Careful analysis of crystallography at 2.4Å resolution with sufficient statistics as well as oocyte assay determine a cytoplasmic chloride binding site and a candidate regulatory site. Comparison between closely related neuron GLUT3 reveals a remarkable role of SP motif in transport regulation, and explains at least in part, the difference of the apparent glucose affinities between GLUT1 and GLUT3 by regulating their conformational equilibrium. Given these advances, this reviewer agrees that this paper will be of specific interest to the field of membrane transporters and of general interest to the much larger fields of membrane transport mechanisms.

We thank the reviewer for his support for the presented work.

In the manuscript, effect of SP-A interaction is carefully investigated by either structural comparison with other GLUT transporters, and by the mutagenesis studies on two isoforms GLUT1 and GLUT3. However, it is unclear how SP-A network allosterically modulate the conformation of whole molecule. Regulatory sites (SP-A) in the N and C domain seems far from the canonical transport site which is located at the interface between two lobes. It is helpful for readers if authors describe how the changes occurred in the SP-A site transmit and affect to the whole transporter structure.

The effect of the SP-A network is more directly on conformational change of the whole molecule, and does directly change the substrate binding site as the reviewer comments. The effect of the A motif on MFS conformational change has been analyzed and discussed in several papers (E.g Martens et al 2018), cf also. line 165f. In the revised manuscript, we have added a more careful discussion of how the SP and A motifs modulates GLUT's conformation lines, cf lines 181-194 .

The term "affinity" should be defined correctly. Determined "affinity" from the Michalis-Menten fitting of the glucose transport is "apparent affinity", as this measurement does not determine the direct binding of glucose to the transporter.

We have adjusted the text to be more precise in our choice of words.

Careful proof reading is recommended, especially about the description of K_m and V_{max} . "K" and "V" should be in Italic, and "m" and "max" should be in subscript.

We have adjusted the manuscript accordingly.

L219

A motif

We are unsure what the reviewer meant by this comment. Perhaps it is a remnant of a copy-paste operation? Residue 219 is not a leucine in GLUT1 but an asparagine. It is not part of the A motif which is located at position 84-93 and 325-334.

Fig2D

It is better for reader to change the color of C-domain SP motif to discriminate C-domain A motif.

We have corrected this. The color of C-domain has been changed to green in the new figure.

January 12, 2021

RE: Life Science Alliance Manuscript #LSA-2020-00858R-A

Dr. Bjørn Panyella Pedersen
Aarhus University
Department of Molecular Biology and Genetics
Gustav Wieds Vej 10
MBG-AU
Aarhus C, Danmark 8000
Denmark

Dear Dr. Pedersen,

Thank you for submitting your revised manuscript entitled "Structural comparison of GLUT1 to GLUT3 reveal transport regulation mechanism in Sugar Porter family". We would be happy to publish your paper in Life Science Alliance pending final revisions necessary to meet our formatting guidelines.

Along with the points listed below, please also attend to the following,

- please add a callout for Figure S6A in your main manuscript text
- please correct Fig 5B and 5C as pointed out by Reviewer 3 in their comments below

A. FINAL FILES:

-- Summary blurb (enter in submission system): A short text summarizing in a single sentence the study (max. 200 characters including spaces). This text is used in conjunction with the titles of papers, hence should be informative and complementary to the title. It should describe the context

and significance of the findings for a general readership; it should be written in the present tense and refer to the work in the third person. Author names should not be mentioned.

B. MANUSCRIPT ORGANIZATION AND FORMATTING:

Sincerely,

Shachi Bhatt, Ph.D.
Executive Editor
Life Science Alliance
<https://www.lsjournal.org/>
Tweet @SciBhatt @LSAJournal

Reviewer #1 (Comments to the Authors (Required)):

The binding of Cl⁻ in the crystal structure was not in question. Whether it was specific binding (i.e., well recognized by the protein residues) and functionally relevant was the issue. Some functional

evidence for the role of Cl⁻ would have supported the authors' interpretation.

The authors' interpretation of the Cl⁻ and the second glucose binding sites is still unconvincing, but the work deserves to be published. It is very challenging to determine the 3D structure of glucose transporters, and we need as much structural information as possible to better understand these important proteins.

I recommend the publication of the manuscript without further modification.

Reviewer #2 (Comments to the Authors (Required)):

This paper indicates that interactions between a cytosolic SP motif and a conserved cytosolic A motif present in the endofacial links stabilize the outward conformational state of GLUTs and thereby increase substrate apparent affinity. A Cl⁻ site in GLUT1 and an endofacial lipid/glucose binding site seem to modulate GLUT1 kinetics to raise the K_m towards glucose. The results provide a possible explanation for the difference between GLUT1 and GLUT3 glucose affinity.

The figures are much improved from the previous version and the paper reads well.

The experimental results provided fit the authors' interesting and attractive hypothesis, although I must say the conclusions derived seem a little tenuous based as they are on small differences between the K_ms of GLUT1 and GLUT3. It is a pity the authors seem unwilling to test their hypothesis more rigorously by doing the Cl⁻ replacement experiment which was suggested. Also, it is very evident, since they are reliant on the hypothetical transient contacts between the SP and A motifs to stabilize the outward facing posture that a molecular dynamic analysis is very much needed to ratify their hypothesis. Anion replacements in silico are certainly rather simple to do and might prove instructive. This will have to wait. The competitive inhibition studies whilst interesting, in my opinion do not really help except to corroborate that GLUT1 and 3 have similar specificities.

Reviewer #3 (Comments to the Authors (Required)):

The resubmission by Tania et al is improved to accommodate reviewers' request. The authors more clearly focus on SP-A network and Cl⁻ site. Now possible uncertainties (the physiological relevance of cytosolic NG and Cl⁻) are clearly discussed and stated. Providing electron density maps of NG, Cl⁻ and its heavy atom congener Br⁻ allow readers to assess its significance.

Figure 5 panels B,C are obviously wrong. Authors discuss GLUT1 K456R and GLUT3 R454K in the text. But data for E454Q and E452Q are shown in the figure. These must be replaced to K456R and R454K ones, which were included in the previous version of the manuscript.

January 19, 2021

RE: Life Science Alliance Manuscript #LSA-2020-00858RR

Dr. Bjørn Panyella Pedersen
Aarhus University
Department of Molecular Biology and Genetics
Gustav Wieds Vej 10
MBG-AU
Aarhus C, Danmark 8000
Denmark

Dear Dr. Pedersen,

Thank you for submitting your Research Article entitled "Structural comparison of GLUT1 to GLUT3 reveal transport regulation mechanism in Sugar Porter family". It is a pleasure to let you know that your manuscript is now accepted for publication in Life Science Alliance. Congratulations on this interesting work.

*****IMPORTANT:** If you will be unreachable at any time, please provide us with the email address of an alternate author. Failure to respond to routine queries may lead to unavoidable delays in publication.*******

DISTRIBUTION OF MATERIALS:

Again, congratulations on a very nice paper. I hope you found the review process to be constructive and are pleased with how the manuscript was handled editorially. We look forward to future exciting

submissions from your lab.

Sincerely,

Shachi Bhatt, Ph.D.

Executive Editor

Life Science Alliance

<https://www.lsjournal.org/>
